# MicroRNAs’ Role in Diagnosis and Treatment of Subarachnoid Hemorrhage

**DOI:** 10.3390/diseases11020077

**Published:** 2023-05-23

**Authors:** Zahra Hasanpour Segherlou, Lennon Saldarriaga, Esaan Azizi, Kim-Anh Vo, Ramya Reddy, Mohammad Reza Hosseini Siyanaki, Brandon Lucke-Wold

**Affiliations:** 1Department of Neurosurgery, College of Medicine, University of Florida, Gainesville, FL 32661, USA; 2College of Medicine, University of Florida, Gainesville, FL 32661, USA

**Keywords:** aneurysmal subarachnoid hemorrhage, microRNAs, exosomes

## Abstract

Subarachnoid hemorrhage (SAH) is most commonly seen in patients over 55 years of age and often results in a loss of many productive years. SAH has a high mortality rate, and survivors often suffer from early and secondary brain injuries. Understanding the pathophysiology of the SAH is crucial in identifying potential therapeutic agents. One promising target for the diagnosis and prognosis of SAH is circulating microRNAs, which regulate gene expression and are involved in various physiological and pathological processes. In this review, we discuss the potential of microRNAs as a target for diagnosis, treatment, and prognosis in SAH.

## 1. Subarachnoid Hemorrhage

Subarachnoid hemorrhage (SAH) typically affects patients over the age of 55, resulting in a significant loss of productivity. In 85% of cases, SAH is caused by the rupture of intracranial aneurysms (IA) which occur due to abnormal dilation of arteries resulting from increased pressure in the arteries and vessel structure disorders. Aneurysms often form at the bifurcation of arteries where the high flow of blood can damage the weakened wall of the artery [1]. While there has been a 17% increase in survival from aneurysmal subarachnoid hemorrhage, survivors commonly experience cognitive impairments that that can significantly impact their daily functioning, quality of life, and working capacity [2]. Not-traumatic SAH can lead to early and secondary brain injuries, with early brain injury occurring within 72 h of symptom onset [2] and secondary brain injury caused by cerebral vasospasm and delayed cerebral ischemia [3]. Approximately 50–90% of patients with angiography experience vasospasm [4]. Therefore, further understanding of the role of neuroinflammation in the pathophysiology of SAH is crucial to identify diagnostic markers and targets for therapeutic intervention [5].

## 2. MicroRNA

MicroRNAs (miRNAs) were discovered about 30 years ago in the nematode *Caenorhabditis elegans* [6]. At the same time, RNA interference pathways were discovered, and the most important one was the 21 nucleotide RNA triggers of silencing machinery. Further research showed that these two pathways are the same gene silencing pathway [7]. More than 2000 miRNAs have been discovered in humans, and it is believed that all of them participate in the regulation of one-third of the genes in the genome [7]. miRNAs are endogenous non-coding RNAs with 18–22 nucleotides. miRNAs interfere with the non-translatable 3′ (3′UTR) regions of the mRNAs and regulate gene expression at the post-transcriptional level. The importance of miRNAs was demonstrated by knocking out genes of the enzyme Dicer and Drosha (two enzymes that have critical function in miRNAs processing); knockout of these genes in the mouse model resulted in embryonic lethality [8,9]. In the same way, any tissue-specific knockout of these genes causes defects in the tissue development [10]. The miRNA gene can be in the introns or exons or can be as standalone transcription units [11,12,13]. Their genes are not usually in the exons because their excision would lead to non-functional protein production [7]. Recent studies have shown that miRNAs are highly conserved in humans [14]. miRNAs have a prominent role in the cellular development and in the nervous system. They have an important role in neuroplasticity, development of neurons, dendritic spine development, neuronal remodeling, memory formation (in the amygdala), neuronal survival, and other neurobiological processes and diseases, and the expression profile can differ in pathological situations [15,16,17,18,19,20]. miRNAs regulate gene expression and are involved in different physiological and pathological processes. miRNAs are tissue-specific; for example, miR-9, miR-124a/b, miR-135, miR-153, miR-183, and miR-219 are expressed in differentiating neurons [21].

Neuroinflammation drives damage progression in IA and SAH. Because of its role in immune cell response regulation and inflammatory gene expression, miRNA could be a promising target for minimally invasive diagnostic and prophylactic purposes [22]. Tissue cells secrete miRNAs into the circulation and other biological fluids inside vesicles. miRNAs can be detected in the cells, tissues, and body fluids such as serum, plasma, tears, urine, or cerebrospinal fluid (CSF) [23]. For this reason, these circulating miRNAs are a novel target for the diagnosis and prognosis of a SAH [24]. Understanding the complete miRNA pathway is important because every miRNA regulates the expression of hundreds of genes [25].

The detection of miRNAs can be accomplished through several methods, including quantitative PCR (qPCR), in situ hybridization, microarrays, and RNA-sequencing [26]. 

## 3. Methods

A narrative review was conducted to explore the current literature regarding the relationship between microRNAs (miRNAs) and subarachnoid hemorrhage (SAH), specifically focusing on the potential use of miRNAs as diagnostic, prognostic, and therapeutic biomarkers. The key words used to search the relevant literature included subarachnoid hemorrhage, aneurysmal subarachnoid hemorrhage, microRNAs, brain aneurysm, and miRNA. A comprehensive literature search was conducted in PubMed, Embase, and Web of Science databases with a date before September 2022. The search strategy included a combination of keywords such as “subarachnoid hemorrhage”, “aneurysmal subarachnoid hemorrhage”, “microRNAs”, “brain aneurysm”, and “miRNA”. The inclusion criteria for the literature review were (1) studies on preclinical models or human subjects, (2) studies in English language, and (3) studies on the association between miRNAs and SAH. 

## 4. microRNA and SAH Diagnosis

A genomic investigation conducted on plasma samples from humans has demonstrated that the levels of circulating miRNAs undergo unique alterations in various disease conditions such as cancer, diabetes mellitus, hypertension, myocardial infarction, and heart failure. Hence, they have the potential to function as disease-specific biomarkers [27,28]. Circulating MiRNAs are modified and secreted into these fluids within extracellular vesicles or are bound by proteins that guard them against degradation [24]. 

Blood biomarkers with high sensitivity and specificity can potentially be useful as a screening tool to detect asymptomatic IAs. In some research, circulating miRNAs have been utilized as novel biomarkers to diagnose the possibility of IAs occurrence in high-risk populations. In a study by Li et al., they discovered that 20 plasma miRNAs were significantly altered in IA patients, irrespective of the complication of IA (such as SAH). Their findings showed that only hypertension and the levels of miR-16 and miR-25 (increased by approximately 1.5) and not age, sex, smoking, or medication were independent predictors for the presence of IAs. These findings indicated that these two biomarkers may be useful biological markers to assess the risk of IAs [29].

In other clinical studies, at least 15 circulating miRNAs have been identified as potential diagnostical biomarkers in SAH or IA. These include miR-1297, miR-502-5p, miR-4320, miR-143, miR-145, miR-155, miR-29a, miR-200a-3p, miR-let7-b, miR-16, miR-25, miR-15a-5p, miR-146-5p, miR-126, and miR-132-3p, among others. There is significant statistical correlation between the up- or downregulation of miRNAs and the severity of SAH [29,30,31,32,33,34,35,36,37,38,39,40]. Plasma miRNA profiling with qRT-PCR further confirms and validates distinct differences in patients with SAH and healthy controls with miR-15a-5p, miR-34a-5p, miR-374a-5p being upregulated and miR-146a-5p, miR-376c-3p, miR-18b-5p, miR-24-3p, and miR-27b-3p being downregulated. The predictability of the patients with SAH is 0.865, and for healthy controls, it is 0.995 [38]. These clinical trials used patient serum or plasma and qRT-PCR to profile and analyze the miRNA, further proving the predictability and accessibility of these miRNA as biomarkers for SAH or IA. In CSF analysis of patients with SAH, miR-204-5p, miR-223-3p, miR-337-5p, miR-451a, miR-489, miR-508-3p, miR-514-3p, miR-516-5p, miR-548 m, miR-599, miR-937, miR-1224-3p, and miR-1301 were different from those in non-SAH groups [41].

## 5. microRNA-Based Therapies for SAH

At present, there is no effective treatment for SAH, but further miRNA profiling would allow for improved neuroinflammation management in patients with IA and SAH. Distinct changes in miRNA in patients after SAH compared to healthy individuals have been reported [42]. Both the increasing and decreasing of miRNAs have been used as a treatment of SAH. 

In preclinical studies, miRNAs have been investigated as potential therapeutic agents and biomarkers for SAH or IA. In a murine SAH model, upregulation of miR-452-3p expression was observed along with increased pro-inflammatory factors and decreased anti-inflammatory factors. The inhibition of miR-452-3p reversed these trends by targeting histone deacetylase 3 (HDAC3). SAH also upregulated p65 acetylation, which was decreased by miR-452-3p inhibitor, leading to the upregulation of IκBα. However, Suberoylanilide hydroxamic acid (SAHA) reversed the protective effect of miR-452-3p inhibitor and aggravated mice brain injury. These findings highlight the potential effect of miR-452-3p and its inhibitor as therapeutic targets for SAH management [43].

Lai et al. discovered miR-193b-3p, a miRNA derived from bone mesenchymal stem cells, in an SAH model with male mice [44]. Systemic injection of miR-193b-3p downregulated HDAC3 and decreased p65 acetylation. Treatment with miR-193b-3p also reduced the levels of inflammatory cytokinesIL-1β, IL-6, and TNF-α in the brain tissue of mice following SAH [44]. These findings suggest that miRNAs and anti-miRNAs can modulate neuroinflammation through the HDAC3/NF-κB signaling in IA, early brain injury, and SAH (Table 1). In another study, Lou et al. demonstrated that the HDAC inhibitor SAHA protected against neuronal injury following SAH by increasing miR-340, which attenuated pyroptosis and the NEK/NLRP3 pathway [45].

In a rat SAH model, miR-103-3p was found to be upregulated and caused a decrease in Cav-1, leading to reduced neuroprotective effects. Therefore, inhibition of miR-103-3p could be a potential therapeutic strategy to preserve Cav-1 and maintain blood–brain barrier integrity, making it a novel target for SAH treatment (Table 1) [46].

Research has demonstrated a significant association between miRNAs and the regulation of NF-κB, both in pro- and anti-inflammatory contexts. Chen et al. investigated the regulation and delivery of miR-124, the most abundant miRNA in the central nervous system, by CX3CL1 and CX3CR1 [47]. Upregulation of miR-124 in microglia inhibits CEBPα, a target protein, and downregulates TNF-α, thereby reducing microglia activation and signaling downstream cascades after SAH [47]. The study suggested that CX3CL1/CX3CR1-mediated transport of miR-124 in exosomes from neurons to microglia may regulate neuroinflammation in an SAH rat model [47]. 

MiRNA-24 targets the 3′UTR of endothelial nitric oxide synthase (NOS3), and elevated miRNA-24 levels have been associated with vasospasm in SAH patients [48]. Conversely, downregulation of miRNA-24 increases HMOX1 expression, resulting in inflammation reduction and improvement in neurological function in a rat SAH model [49]. 

MiRNA changes can also affect the brain-derived neurotrophic factor (BDNF)/tyrosine kinase B (TrkB)/cAMP-response element-binding protein (CREB) (BDNF/TrkB/CREB) pathway [22]. In a rat SAH model, Zhao et al. targeted BDNF with miR-206 delivered through exosomes derived from human umbilical cord mesenchymal stem cells (hucMSCs) [50]. Knockdown or down regulation of miR-206 increased BDNF expression in rats with SAH through the CREB pathway in vivo, resulting in improved neurological function [51]. CREB is a target of miR-34b, and downstream activation of the PI3K/Akt/NF-κB pathway is known to be influenced by phosphorylated CREB, leading to inhibition of NF-κB activation and a reduction in the proinflammatory response [52].

MiR-140-5p has demonstrated neuroprotective properties by suppressing toll-like receptor 4 (TLR4) and inhibiting downstream phosphatidylinositol 3-kinase/AK/nuclear factor-κB (PI3K/AKT/NF-κB) inflammatory signaling in rat brain tissue. A study showed that microglia-secreted extracellular vesicles (microglia-EVs) inhibited microglia activation and decreased TNF-α and IL-1β release after injection of miR-140-5p. Microglia-EVs were able to transfer miR-140-5p into microglia. Treating with microglia-EVs-miR-140-5p also reduced macrophage differentiation-associated (MMD) and blocked the inflammatory cascade and microglia response in SAH rats by suppressing the PI3K/AKT and Erk1/2 pathway [53,54]. Furthermore, increased miR-140-5p has been found to downregulate activin-like kinase 5 (ALK5) and NADPH oxidase 2 (NOX2), consequently inhibiting inflammatory M1 microglia activation in SAH mice [55]. These findings suggest that miR-140-5p may have therapeutic potential for the treatment of neuroinflammatory disorders such as SAH.

Makino et al. demonstrated in an aneurysm mouse model that tetracycline derivatives, including minocycline and doxycycline, have anti-inflammatory effects that could be used in aneurysm stabilization and rupture prevention [56]. Both minocycline and doxycycline treatments, through intraperitoneal injection and gavage, respectively, were found to have beneficial effects compared to their corresponding sham groups. While Makino et al. did not include a mechanistic investigation, subsequent studies have found that minocycline and doxycycline enhance brain-derived neurotrophic factor (BDNF) expression, decrease reactive oxygen production, and lessen inflammation through regulation of miR-155 and miR-210 [57,58]. These findings suggest that miR-155 and miR-210 may have therapeutic potential in the prevention of aneurysm rupture through their anti-inflammatory effects. In a murine SAH model, miR-22 was found to be upregulated compared to control mice without SAH, resulting in a decrease in IL-6 [59]. Lowering the expression of miR-22 increased IL-6 expression and led to neuroprotective effects. Increasing the miR-22 expression also suppressed the caspase-3/Bax signaling pathway. These results suggest that miR-22 may be a potential therapeutic agent for the treatment of SAH [59]. 

## 6. Exosomes

Recent scientific research has shed light on the potential therapeutic application of exosomes in the treatment of SAH. Exosomes are extracellular nanovesicles (30–120 nm) enclosed in lipid membrane secreted by multiple cell types and contain various cargo molecules including miRNAs, proteins, and lipids [60,61]. Although previously thought to eliminate non-functional proteins in cells, recent research suggests that exosomes play a significant role in intercellular communication by transferring and regulating immune responses to neighboring cells [62]. Furthermore, miRNAs delivered by exosomes have been shown to be more than those in the cell cytoplasm and are less likely to degrade, making them an optimal delivery agent in preclinical studies [30]. Several preclinical studies have specifically investigated the potential therapeutic role of exosomal miRNAs in SAH, which is further discussed below.

It is well documented that DNA-binding protein 43 (TDP-43) mis-localization is widely known to lead motor neuron death [63]. In addition, TDP-43 levels in cerebrospinal fluid (CSF) have shown promise as a prognostic biomarker for SAH, indicating its association with response to neuronal injury [64]. Additionally, studies have reported low levels of miR-140-5p in rat model of intracerebral hemorrhage [53]. Based on these evidences, the recent scientific literature has highlighted the potential therapeutic implications of exosomes derived from adipose tissue-originated stromal cells (ADSCs) in regenerative medicine. This study aimed to investigate the potential protective effects of ADSC-Exosomes, which contain miR-140-5p, on neuronal injury caused by SAH in a rat model. The study found that ADSC-Exosomes could protect against neuronal injury caused by TDP-43 by promoting cell viability and suppressing cell apoptosis. This study demonstrated that ADSC-Exososome-miR-140-5p could prevent TDP-43-induced neuronal injury and attenuate neurological dysfunction of SAH rats by inhibiting insulin-like growth factor binding protein 5 (IGFBP5) and activating the PI3K/Akt signaling pathway [65]. 

One study used bone marrow MSC-derived exosomes (BMSC-Exos) to deliver miR-193b-3p in a mouse model of SAH, which suppressed the activity of histone deacetylase 3 (HDAC3) and led to the acetylation of nuclear factor kappa-light-chain-enhancer of activated B cells (NF-kB) p65, ultimately attenuating neuroinflammation in EBI [44]. Moreover, another study showed that BMSC-Exos-delivering miR-129-5p in a SAH rat model led to anti-inflammatory and antiapoptotic effects by attenuating the HMGB1-TLR4 pathway [66]. In addition, a study utilized MSC-derived exosomes to delivermiR-21, which reduced neuronal apoptosis and alleviated SAH-induced cognitive dysfunction by promoting neuronal survival and alleviating EBI after SAH via the PTEN/Akt pathway [67]. Further evidence showed that when miR-21 was knocked out or a PTEN/Akt inhibitor was administered (MK2206), MSC-EV could not suppress EBI and neuronal apoptosis induced by SAH [67].

In addition to MSC-derived exosomes, a recent study also utilized exosomes derived from human umbilical cord mesenchymal stem cells (hubMSCs). Oxy hemoglobin (OxyHb)-treated PC12 cells were transfected with hubMSCs-exosomes alone or with miR-26b-5p inhibitor [68]. The inhibitor abolished any promoting effects of exosomes on PC12 cell proliferation and cell apoptosis. Further experiments used pcDNA- methionine adenosyltransferase II alpha (MAT2A), which had the same effect as miR-26b-5P inhibitor. In addition, injecting miR-26b-5p inhibitors resulted in increased MAT2A protein expression, increased inflammatory mediators, and aggravated neurological symptoms in SAH rat models [68]. These results suggest that the target gene of miR-26b-5p may be the MAT2A gene.

Lastly, another study showed that after SAH, the delivery of exosomal miR-124 from neurons to microglia was reduced, while there was an increase in C/EBPα expression [47]. This increase in C/EBPα expression was due to CX3CL1/CX3CR1 overexpression. Several experiments in the study detailed that the CX3CL1/CX3CR1 axis might have a protective effect after SAH by promoting miR-124 transport from neurons to microglia, which can attenuate microglial activation and neuroinflammation [47]. Therefore, this study suggests that the CX3CL1/CX3CR1 axis could also be a potential therapeutic target that affects downstream miRNA expression, ultimately inhibiting early brain injury after SAH [47]. 

In a rat model of SAH, Cheng et al. studied the effect of mesenchymal stem cell-derived extracellular vesicles. They found that KLF3-AS1, delivered by bone marrow mesenchymal stem cell-derived extracellular vesicles, upregulates TCF7L2 expression by binding to miR-138-5p. This ultimately led to a decrease in neurological dysfunction and endothelial damage after SAH [69]. Zhou et al. examined the role of the long noncoding RNA metastasis-associated lung adenocarcinoma transcript 1 (MALAT1) in a SAH mouse model and in vitro. They found that MALAT1 expression was increased in the brains of mice and in vitro SAH model. Knocking down the MALAT1 gene decreased neuronal apoptosis and reduced the production of reactive oxygen species by neurons. They showed that MALAT1 was associated with miR-499-5p/SOX6 axis [70]. Cai et al. conducted a study on the role of circARF3 in the SAH rat model and found that upregulating circARF3 improved the integrity of the blood–brain barrier and neurological function while reducing the apoptosis of neurons and microglia in the ipsilateral basal cortex. These effects were shown to be regulated by the miR-31-5p-activated MyD88-NF- κB pathway [71]. In a separate study, Ru et al. demonstrated that miR-706 attenuates white matter injury in SAH mice model via the PKCa/MST1/NF-κB pathway and the release of inflammatory cytokines [72]. Yu et al. reported that p53/microRNA-22 had neuroprotective effects in the SAH mice model by regulating IL-6 mRNA expression and the caspase-3/Bax signaling pathway [59]. Similarly, Yang et al. showed that treating SAH with melatonin increased protection against early brain injury through the H19-miR-675-P53-apoptosis and H19-let-7a-NGF-apoptosis pathways in their SAH mice model study [73]. 

Bhimani et al. discovered that miRNAs had both anti-inflammatory and anti-apoptotic effects, and that these effects were mediated through the CREB and PI3K/Akt/NF-κB pathways. They also suggested that delivering miRNAs through exosomes could be a potential treatment for vasospasm [22].

In conclusion, several preclinical studies have shown promising results in attenuating EBI, neuroinflammation, and neuronal apoptosis by targeting or delivering various miRNAs via exosomes from different sources. However, further analysis and experimentation are necessary to determine which specific miRNAs target specific pathways, potential side-effects, the optimal mode of delivery, and other factors before this research can be translated into clinical trials. Currently, clinical trials are studying the expression panel of various miRNAs in the context of SAH or intracranial aneurysms in general, and they have shown promise with various miRNAs being upregulated or downregulated with high specificity in such settings. Because miRNAs can be detected earlier than proteins, which manifest in CSF or blood later in the progression of tissue injury, they may be more suitable as potential clinical biomarkers [30].

## 7. microRNA and SAH Prognosis

MicroRNA levels may serve as a biomarker for prognosis in patients with SAH [30]. For instance, elevated levels of the microRNAs let-7b-5p, miR-19b-3p, miR-125-5p, miR-221-3p, miR-21-5p, and miR-27a-3p in the CSF have been linked to a higher likelihood of delayed cerebral vasospasm in patients with aneurysmal SAH (Table 1). However, it is important to note that these changes do not necessarily occur at the same levels or magnitudes in blood plasma. In plasma, several miRNAs, including let-7a-5p, miR-146a-5p, miR-204-5p, miR-221-3p, miR-23a-3p, and miR-497-5p, were elevated three days after aneurysmal SAH and correlated with delayed cerebral vasospasm. This represents a different group of microRNA indicators compared to those in CSF. It is worth noting that these plasma samples became undetectable by the seventh day possibly due to the breakdown of molecules over time to levels that are no longer detectable [74].

Cerebral vasospasms is a major cause of death and poor outcomes in patients with SAH. Studies have shown that certain miRNAs including miR-337-5p, miR-519b-3p, and miR-548m are significantly altered inpatients who experience cerebral vasospasm. MiRNAs play a crucial role in controlling gene expression, including genes involved in SAH, which can lead to both overexpression and under expression of target genes [41]. Studies have also shown that low expression of miR-195-5p is linked to cerebral vasospasm and poor outcomes in SAH patients [75]. In other studies, low levels of miR-630 in the CSF were associated with better endothelial function after SAH [76]. In a study comparing the miRNA levels in CSF between SAH patients who developed cerebral vasospasm and those who did not, it was found that miR-27a-3p, miR-516a-5p, miR-566, and miR-1197 were significantly different between the two groups. This suggests that analyzing miRNA profile can predict which patients are at the higher risk of developing cerebral vasospasm following SAH [41]. Since miRNAs can bind to genes that control the expression of several different proteins, it is believed that these molecules can affect the brain’s healing response. Clinical results suggest that some miRNA molecules decrease in quantity rapidly after SAH because they may contribute to brain injury, while others may also increase in value because they are released from the brain in response to the trauma. The changes in these molecules may serve as an essential biomarker for assessing how the brain is healing itself after traumatic neurological injury [77]. Since miRNA levels change drastically from day zero to day nine time points in patients suffering from SAH, it is important to study these specific changes in more depth to identify biomarkers leading to poor prognosis. The blood–brain barrier may also explain why miRNA levels in the CSF differ from those in the blood. Depending on the molecules that are actively transported across this barrier, different levels may be observed in each respective region [78]. One challenge in studying the effects miRNAs on SAH is the small sample size often used in these studies. With a larger sample size, more robust evidence may be obtained [79]. 

## 8. Conclusions

While some studies have identified specific miRNAs that may be upregulated or downregulated in SAH patients, the specific mechanism by which these miRNAs contribute to SAH remain unclear. Some studies have suggested that miRNAs may be useful targets for therapies aimed at reducing inflammation or promoting neuronal survival after SAH. Some studies have suggested that miRNA profiles in cerebrospinal fluid or blood may be useful in predicting outcomes or identifying patients at high risk for cerebral vasospasm. Although several studies have explored the potential role of miRNAs in SAH, there is still much to be learned about their specific mechanism and their potential as diagnostic and therapeutic targets.

## Figures and Tables

**Table 1 diseases-11-00077-t001:** Micro-RNAs role in the diagnosis, treatment and prognosis of SAH.

First Author	Year	miRNA(s) Evaluated	Subjects Evaluated	Specimen Evaluated	Main Findings
Su XW	2015	miR-132-3p, miR-324-3p	Human	CSF	Circulating miR-132-3p and miR-324-3p may be potential biomarkers for acute aneurysmal SAH.
Wang WH	2016	miR-29a	Human	Blood	miR-29a may be a potential biomarker in the development of intracranial aneurysm.
Zaccagnini G	2017	miR-210	Mouse	Ischemic tissue	Overexpression and significance in ischemic tissue damage.
Sheng B	2018	miR-1297	Human	Serum	Early serum miR-1297 is an indicator of poor neurological outcome in patients with aSAH.
Sheng B	2018	miR-502-5p	Human	Serum	Persistent high levels of miR-502-5p are associated with poor neurologic outcome in patients with aneurysmal subarachnoid hemorrhage.
Feng X	2018	miR-143, miR-145	Human	Serum	Lower miR-143/145 levels and higher MMP-9 levels may be associated with intracranial aneurysm formation and rupture.
Li	2018	miR-24	Rat	Brain tissue	Upregulation of miR-24 expression led to vasospasm by suppressing endothelial nitric oxide synthase expression after SAH.
Yu S	2018	miR-22	Rat	Brain tissue	Neuroprotective effects in regulating inflammation and apoptosis.
Yang X	2019	miR-155	Human	Blood	A functional polymorphism in the promoter region of miR-155 predicts the risk of intracranial hemorrhage caused by ruptured intracranial aneurysm.
Zhao	2019	miR-206	Rat	Used as a therapeutic target	HucMSCs-derived miR-206-knockdown exosomes targeted BDNF, contributing to neuroprotection after SAH.
Wang S	2019	miR-140-5p	Rat	Used as a therapeutic target	Attenuated neuroinflammation and brain injury by targeting TLR4.
Geng W	2019	miRNA-126	Rat	Used as a therapeutic target	Exosomes from miRNA-126-modified ADSCs promote functional recovery after stroke in rats by improving neurogenesis and suppressing microglia activation.
Yang F	2020	miR-126	Human umbilical vein endothelial cell	Human umbilical vein endothelial cell	miR-126 may be involved in the development and rupture of intracranial aneurysms.
Lai	2020	miR-193b-3p	Mouse	Used as a therapeutic target	Systemic exosomal delivery of miR-193b-3p attenuated neuroinflammation and improved neurological function after SAH.
Chen	2020	miR-124	Rat	Used as a therapeutic target	CX3CL1/CX3CR1 axis promoted exosomal delivery of miR-124 from neuron to microglia, attenuating early brain injury after SAH.
Xiong L	2020	miRNA-129-5p	Rat	Used as a therapeutic target	Exosomes from bone marrow mesenchymal stem cells can alleviate early brain injury after subarachnoid hemorrhage through miRNA129-5p-HMGB1 pathway.
Gao X	2020	miRNA-21-5p	Rat	Used as a therapeutic target	Extracellular vesicle-mediated transfer of miR-21-5p from mesenchymal stromal cells to neurons alleviates early brain injury to improve cognitive function via the PTEN/Akt pathway after subarachnoid hemorrhage.
Wang	2021	miR-103-3p	Rat	Used as a therapeutic target	Inhibition of miR-103-3p preserved neurovascular integrity by upregulating caveolin-1 expression after SAH.
Deng	2021	miR-24	Rat	Used as a therapeutic target	miR-24 regulated inflammation and neurofunction by targeting HMOX1 expression in rats with cerebral vasospasm after SAH.
Liu Z	2021	miRNA-26b-5p	Rat	Used as a therapeutic target	MiR-26b-5p-modified hUB-MSCs-derived exosomes attenuate early brain injury during subarachnoid hemorrhage via MAT2A-mediated p38 MAPK/STAT3 signaling pathway.
Cai L	2021	circARF3	Rat	Used as a therapeutic target	Up-regulation of circARF3 reduces blood-brain barrier damage in rat subarachnoid hemorrhage model via miR-31-5p/MyD88/NF-κB axis.
Ru X	2021	miRNA-706	Mouse	Used as a therapeutic target	MiR-706 alleviates white matter injury via downregulating PKCα/MST1/NF-κB pathway after subarachnoid hemorrhage in mice.
Lu	2022	miR-452-3p	Rat	Used as a therapeutic target	miR-452-3p inhibited HDAC3 expression, leading to activation of NF-κB signaling and exacerbation of early brain injury after SAH.
Qian Y	2022	miR-140-5p	Mouse	Used as a therapeutic target	Alleviated M1 microglial activation in brain injury via miR-140-5p delivery.
Wang P	2022	miRNA-140-5p	Rat	Used as a therapeutic target	Exosome-encapsulated microRNA-140-5p alleviates neuronal injury following subarachnoid hemorrhage by regulating IGFBP5-mediated PI3K/AKT signaling pathway.
Cheng M	2022	miRNA-83-5p	Rat	Used as a therapeutic target	Extracellular vesicles derived from bone marrow mesenchymal stem cells alleviate neurological deficit and endothelial cell dysfunction after subarachnoid hemorrhage via the KLF3-AS1/miR-83-5p/TCF7L2 axis.
Zhou X	2022	miRNA-499-5p	Rat	Used as a therapeutic target	Suppression of MALAT1 alleviates neurocyte apoptosis and reactive oxygen species production through the miR-499-5p/SOX6 axis in subarachnoid hemorrhage.
Luo	2023	miR-340	Rat	Used as a therapeutic target	HDAC inhibitor SAHA upregulated miR-340 expression, which inhibited NEK7 signaling and attenuated pyroptosis after SAH.
Wang P	2023	miR-140-5p	Rat	Used as a therapeutic target	Attenuated microglia activation and inflammatory response via MMD downregulation.

CSF: cerebrospinal fluid, SAH: subarachnoid hemorrhage, MMP: Matrix metalloproteinase, BDNF: Brain derived neurotrophic factor, ADSC: Adipose derived stem cells.

## Data Availability

Not applicable.

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
