# Peer review of "MicroRNAs’ Role in Diagnosis and Treatment of Subarachnoid Hemorrhage"

_diseases, 2023, doi:10.3390/diseases11020077_

Round 1
Reviewer 1 Report (Previous Reviewer 3)
The authors have properly revised the paper according to my previous comments.
The paper still require English revisions.
Author Response
Thank you for your comment. We edited the paper for its English language problems.
Reviewer 2 Report (Previous Reviewer 1)
Some grammatical mistakes were corrected. The idea of this narrative review is that multiple microRNA are up- or down-regulated after the SAH and maybe this might be interesting in the future research. No analysis performed, no discussion or any pathophysiological mechanisms were presented. Conclusions are very abstract, presenting thoughts about the possible application of microRNA in the future, but not concluding what have been found in the present review.
English is improved, few spelling errors still present (e.g. "moue model", line 180)
Author Response
Thank you very much for your comments. We edited the grammatical and spelling errors of the paper. I am afraid to say that based the papers that we have written the manuscript on, we can not provide the pathophysiological pathways for miRNAs. I am so sorry to tell that, but the papers that we have reviewed mostly have focused on their study design and result of their study and no proper pathway has discovered for the miRNAs. As you mentioned in your comment this paper's purpose is to provide which miRNA is up-regulated and which one is down-regulated in SAH, and unfortunately the study in this field is very limited that we can not do analysis or write a proper discussion about it.
Thank you again.
Round 2
Reviewer 2 Report (Previous Reviewer 1)
It seems, that maximal corrections were made as it can be done in this review.
This manuscript is a resubmission of an earlier submission. The following is a list of the peer review reports and author responses from that submission.
Round 1
Reviewer 1 Report
Narrative review of multiple microRNA studies, with no clear idea of the review, multiple grammatical mistakes, no analysis, no discussion and not argumented conclusions.
Reviewer 2 Report
The review article needs to be better written and organized. Several problems throughout the manuscript include spelling mistakes, abbreviations, and no proper order in explaining the miRNAs related to SAH diagnosis, therapeutics, and prognosis. Reading the exosome part detached entirely from the topic. Detailed miRNA biogenesis is unnecessary as no novel findings are associated with SAH. The role of miRNAs in the SAH or IA mechanism needs to be adequately explained. Combine all three tables into a single table with details and proper presentation. Gathering information from other publications instead of presenting a hypothetical model with a figure using available published data could be easy to understand. From their back reference, this review looks rewritten in another format, resembling a recent publication summary.
Reviewer 3 Report
In this paper, the authors review the current evidence on the role of microRNAs (miRNAs) in patients with subarachnoid hemorrhage. While the study is potentially interesting, there are several improvements that should be made before considering the manuscript for publication.
-
The introduction is poorly organized, as some information is too basic (such as how subarachnoid hemorrhage occurs), while other information is too specialized (such as the detailed description of intracellular miRNA metabolism). The authors should revise the introduction to provide a better balance of information for readers.
-
The paper lacks a methods section, which makes it difficult for readers to understand how the search was performed. While the study is a narrative review, the authors should still provide details about the search strategy. This should include information such as the databases searched, the date of the search, the keywords used, the researchers involved in the search and their expertise, and the inclusion and exclusion criteria. These details will help readers to understand the methodology of the study. Moreover, specify in the title that this is a narrative review.
-
The tables in the paper are less informative than they could be. To improve their usefulness, the authors should indicate, for each paper, the first author, the year of publication, the miRNA(s) evaluated, the subjects evaluated (animal or human), the specimen evaluated (CSF, serum, plasma), and the main findings. These details will help readers to better understand the papers included in the review.
-
The English in the manuscript is poor and needs to be revised by a professional English service to ensure clarity and accuracy.
Overall, while the study has potential, these improvements are necessary to ensure that the manuscript is of sufficient quality for publication.